# Strategies for Avoiding Typical Drug–Drug Interactions and Drug-Related Problems in Patients with Vascular Diseases

**DOI:** 10.3390/medicina59040780

**Published:** 2023-04-17

**Authors:** Klaus Peter Schmelzer, Dominik Liebetrau, Wolfgang Kämmerer, Christine Meisinger, Alexander Hyhlik-Dürr

**Affiliations:** 1Pharmacy Department, University of Augsburg, 86156 Augsburg, Germany; 2Vascular Surgery, Medical Faculty, University of Augsburg, 86156 Augsburg, Germany; 3Epidemiology, Medical Faculty, University of Augsburg, 86156 Augsburg, Germany

**Keywords:** medication reconciliation, drug-related side effects, adverse reactions, drug interactions, medication errors, pharmacy service hospital

## Abstract

*Background and objectives*: Drug–drug interactions and drug-related problems in patients with vascular diseases are common. To date, very few studies have focused on these important problems. The aim of the present study is to investigate the most common drug–drug interactions and DRPs in patients with vascular diseases. *Materials and Methods*: The medications of 1322 patients were reviewed manually in the time period from 11/2017 to 11/2018; the medications of 96 patients were entered into a clinical decision support system. Potential drug problems were identified, and a read-through consensus was reached between a clinical pharmacist and a vascular surgeon during the clinical curve visits; possible modifications were implemented. The focus was on additional dose adjustment and drug antagonization on drug interactions. Interactions were classified as contraindicated/high-risk combination (drugs must not be combined), clinically serious (interaction can be potentially life-threatening or have serious, possibly irreversible consequences), or potentially clinically relevant and moderate (interaction can lead to therapeutically relevant consequences). *Results*: A total of 111 interactions were observed. Of these, 6 contraindicated/high-risk combinations, 81 clinically serious interactions, and 24 potentially clinically relevant and moderate interactions were identified. Furthermore, 114 interventions were recorded and categorized. Discontinued use of the drug (36.0%) and drug dose adjustment (35.1%) were the most common interventions. Mostly, antibiotic therapy was continued unnecessarily (10/96; 10.4%), and the adjustment of the dosage to kidney function was overlooked in 40/96; 41.7% of the cases. In the most common cases, a dose reduction was not considered necessary. Here, unadjusted doses of antibiotics were found in 9/96, 9.3% of the cases. Notes for medical professionals summarized information that did not require direct intervention but rather increased attention on the part of the ward doctor. It was usually necessary to monitor laboratory parameters (49/96, 51.0%) or the patients for side effects (17/96, 17.7%), which were expected with the combinations used. *Conclusions*: This study could help identify problematic drug groups and develop prevention strategies for drug-related problems in patients with vascular diseases. A multidisciplinary collaboration between the different professional groups (clinical pharmacists and surgeons) might optimize the medication process. Collaborative care could have a positive impact on therapeutic outcomes and make drug therapy safer for patients with vascular diseases.

## 1. Introduction

Patients with vascular diseases usually suffer from generalized arteriosclerosis as a systemic disease [1]. This is accompanied by several concomitant diseases, such as arterial hypertension, hyperlipidemia, coronary heart disease, arterial occlusive disease, and diabetes mellitus [1]. These various comorbidities are usually treated with a variety of medications. For this reason, patients with vascular diseases are particularly affected by drug–drug interactions (DDIs) and drug-related problems [2]. A drug interaction can affect the effect of a medication or cause unwanted side effects. A drug-related problem (DRP) was defined by the Pharmaceutical Care Network Europe (PCNE) [3] as an event or circumstance involving drug therapy that actually or potentially interferes with desired health outcomes, which mainly includes unnecessary drug treatment, inadequate drug treatment, ineffective drug treatment, adverse drug event, inappropriate dosage, and poor adherence [4]. DDIs and DRPs are generally a major problem [5]. They are responsible for approximately 5% of all hospital admissions, particularly in elderly patients (10–15%). Furthermore, this former study revealed that 5.7% of all inpatients are affected by drug interactions [6]. There is one prior study [2] that focused on these patients and investigated problems, such as medication errors, PIM (potential inadequate drugs e.g., Beers Criteria for older adults [7]), double prescriptions, interactions, unintentional treatment discontinuations, dose frequencies, dose adjustments in renal or hepatic insufficiency, and medication history. Although interactions were recorded, they were not evaluated, and avoidance strategies were not recommended. Similarly, there are original publications showing that DDIs and DRPs led to increased hospitalization rates in patients with cardiovascular disease [8,9]. The present study should add to the current knowledge by identifying the most common DRPs and DDIs in patients with vascular diseases requiring surgical intervention. Furthermore, strategies to avoid these problems will be developed. The results will be summarized in a chart that offers surgeons a systematic approach to increase the safety of their patients. 

Methods Study Population:

The study is based on data collected from patient medical records at the University Hospital of Augsburg from November 2017 to November 2018. During this period, 32 interprofessional curve visits (reviewing the medication documented in patient charts) were performed once a week. The medications of 1322 patients who were in vascular surgery ward at the time and had to undergo vascular surgery were examined. 

Data source

The medications were reviewed by a pharmacist specialized in clinical pharmacy and a vascular surgeon.

Identification of DRPs and DDIs

To identify and analyze the DRPs and DDIs, patient medications were reviewed visually, and the medications of 96 of these patients were entered into a clinical decision support system, (CDSS; AiDKlinik Release 3.5) that also contains an interaction database and other features (see below). Subsequently, the medications of these 96 patients were entered into another interaction database (Lexicomp^®^ Drug Interactions © 2022). This was performed because the interactions and the level of the respective mechanism were to be scientifically described and supported by the literature references. These 96 patients were selected based on the fact that they received a drug from one of the critical Anatomical Therapeutic Chemical (ATC) [10] groups (ATC A12; B01; C09; J01; and M01). In the ATC classification system, the active substances are divided into different groups according to the organ or system on which they act and their therapeutic, pharmacological, and chemical properties. Drugs are classified in groups at five different levels.

Statistical analysis procedure

The DRPs were anonymized, categorized, and manually entered on a spreadsheet for systematic evaluation. In these 96 patients, DRPs were identified (321 DRPs) that required pharmacist intervention.

DRPs include medication errors (errors in prescribing, dispensing, or administering a drug). Additionally, they include adverse drug reactions (harmful and unintended responses to a drug used in a normal dosage) [11]. 

A CDSS is an active knowledge system that uses patient data, including laboratory data, to provide decision support for the treatment of patients [12,13].

The CDSS AiDKlinik Release 3.5 checks the following DRPs:Interactions;Dosing in patients with renal insufficiency, considering the current clearance, which was transferred from the Hospital Information System (HIS);Double medical prescriptions;Potentially inadequate medication for the elderly;Maximum dose.

In addition, the software Lexicomp^®^ Drug Interactions [14] was used to scientifically represent the interactions. There were no major differences in the results of drug interactions between the two systems.

The categorization into the three categories of Contraindicated/High-Risk Combinations, Clinically Serious Interactions, and Potentially Clinically Relevant Moderate Risks was adopted from the analysis of the CDSS and the Lexicomp drug database. The results of the curve visits were discussed with the responsible ward physicians in an interdisciplinary meeting, during which the changes were also discussed. In some cases, the interactions were accepted as a medical necessity for the therapy of the patient. The interventions were documented and evaluated. 

The study was approved by the Ethics Committee of the Ludwig Maximilian University of Munich (Project number: 22-0196; 13 May 2022). Because the data for the study were gathered from existing documents based on routine work and anonymized data, no formal consent by patients was required.

### Statistical Analysis

Data analysis was carried out with IBM^®^ SPSS^®^ Statistics Version 25 and 26 and with Microsoft Office Excel 365. Frequencies are reported as absolute numbers and percentages. Continuous variables are given as median and interquartile ranges or mean and standard deviations as appropriate.

## 2. Results

In this study based on 96 of 1322 older vascular surgery patients, a total of 111 drug interactions were found by using the CDSS. Of these, 6 (5.4%) interactions were classified as contraindicated/high-risk combinations, 81 (77.4%) as clinically serious interactions, and 24 (21.6%) as potentially clinically relevant moderate interactions.

A total of 66 of the 96 patients were male, and 52 (54.2 %) patients suffered from diabetes (2 patients with diabetes type 1). The mean age of the patients was 73.6 years. Twelve (12.5 %) patients had an estimated glomerular filtration rate (eGFR) of less than 30 mL/min. The examined patients received on average of 10.6 different drugs (Table 1).

The main indication for hospitalization was atherosclerosis of the extremity arteries (ICD-10 I70), followed by embolism and thrombosis of the arteries of the lower extremities (ICD-10 I74). Finally, some patients came to hospital for treatment of the vascular complications of their diabetes disease (ICD-10 E11) (Table 1). The most common secondary diagnoses or comorbidities documented were hypertension, diabetes, and peripheral artery occlusive disease PAOD.

### 2.1. Contraindicated/High-Risk Combinations

In four cases, the combination of eplerenone/potassium chloride was detected, and two patients were treated with a combination of new oral anticoagulants (NOACs) and low-molecular-weight heparin (LMWH).

### 2.2. Clinically Serious Interactions

Clinically serious interactions were seen in 81 (73.0%) cases (Table 2). Most frequently (24 times), an interaction between acetylsalicylic acid (ASA) and metamizole was observed. A treatment with nonsteroidal anti-inflammatory agents (NSAIDs) in patients with an eGFR below 60 mL/min was the case in 42 patients (43.8%). In 11 cases, the critical combination of angiotensin-converting enzyme inhibitors, such as ramipril and potassium salts, and potassium-sparing diuretics, such as spironolactone and potassium salts, was found. NSAIDs, such as ibuprofen and salicylates, were combined in seven cases. A combination of oral quinolones, such as ciprofloxacin, with iron preparations, such as iron-II-glycine-sulfate oral iron, for the substitution of an iron deficit was present in two cases.

Other clinically serious interactions seen were the combination of acetylsalicylic acid with venlafaxine or citalopram, a combination of amiodarone and metoprolol, atorvastatin and clarithromycin, and atorvastatin and amiodarone.

### 2.3. Potentially Clinically Relevant Moderate Risks

Potentially clinically relevant moderate risk combinations present in the study sample were prednisolone/hydrochlorothiazide or torasemide, amlodipine/simvastatin, prednisolone/ASA, levothyroxine-natrium/calcium, combinations with vitamin D and/or other drugs, and combination of oxycodone/pregabalin (Table 2). 

### 2.4. Interventions

Altogether, 114 interventions were recorded and categorized (Table 3). In 41 cases (36.0%), it was necessary for a drug to be discontinued. In most cases, an antibiotic was not stopped after sufficient therapy. In 40 cases (35.1%), an adjustment of the dosage was necessary. Often, the dosage was not adjusted to kidney function. In some cases, antibiotics were prescribed with too low a dosage or too rarely. In 13 (11.4%) cases, another substance more suitable for the same indication due to interactions had to be applied. A change of statin was usually necessary. In eight (7%) cases, it was required to apply a drug due to existing interactions. Treatment with a proton pump inhibitor was necessary to avoid gastrointestinal bleeding. In 12 (10.5%) cases, handwritten errors in the patient curve had to be corrected.

Notes for medical professionals (*n* = 96) summarized information for the ward doctor that required increased attention rather than direct intervention (Table 4).

Twenty percent of the patients received potentially inappropriate medication. In 49 (51.0%) cases, the medication given made it necessary to closely monitor certain laboratory values, in particular potassium or serum creatinine, to prevent harm to the patient. 

The ward doctors were advised 17 (17.7%) times to watch out for specific side effects that the drugs used in this combination can trigger. In most cases, it was necessary to watch out for an increased tendency of the patient to bleed. Potentially inappropriate medication was indicated in 18 (18.8%) cases. In most cases, Z-substances or benzodiazepines were used in elderly patients. 

The discontinuation of antibiotic therapy was a common reason why a note to the ward physician was needed. A planned discontinuation date based on the indication would be helpful in this case.

A hint regarding a correct application time was indicated in four (4.2%) cases.

A possible workflow for the structured approach to avoid typical drug-drug interactions and drug-related problems is shown in (Figure 1).

## 3. Discussion

One prior publication dealt with general DRPs [2]; the interactions were recorded but not described in detail. It could be shown that vascular surgery patients are highly impaired by medication errors and DRPs. This finding was confirmed by our study, which also focused on interactions; however, to gain new insights, the interactions were examined in more detail and described comprehensively. Furthermore, strategies that can enable the ward doctor to avoid these interactions were determined. Therefore, this study aims to fill existing gaps and thereby extend current knowledge.

### 3.1. Contraindicated/High-Risk Combinations

In the present study, eplerenone/potassium chloride was used at the same time, a combination that increases the risk of developing life-threatening hyperkalemia. Patients receiving eplerenone for their underlying disease (e.g., heart failure) should not be treated with potassium supplements or potassium-sparing diuretics [15]. Another high-risk combination was the concomitant use of new oral anticoagulants (NOACs) and lowmolecular-weight heparin (LMWH) or unfractured heparin (UFH), a combination that is expected to increase the risk of bleeding [16]. A simultaneous treatment with these two active ingredients is only indicated when changing therapy. Mostly, the discontinuation of NOACs is forgotten in everyday clinical practice.

### 3.2. Clinically Serious Interactions

The most common interaction identified in our study was the clinically severe interaction between ASA and metamizole. Due to the patient clientele, most patients were prescribed oral ASA in a dosage of 100 mg/day for platelet aggregation inhibition (PAI). NSAIDs, such as diclofenac, naproxen, and ibuprofen, should be avoided because of their negative effects on renal function [17,18], in particular if the eGFR is below 60 mL/min [19]. The inhibition of prostaglandin synthesis leads to impaired blood flow to the kidneys and thus to a marked decrease in GFR [18]. The effect is dose-dependent, but the literature explicitly states that no “safe” dose can be defined [18]. Metamizole is usually used orally in a dose of 4 × 1 g daily on the ward that was studied, but AiD-Klinik reports a clinically serious interaction when combined with ASA. Delayed administration (ASA at least 30 min. before metamizole) is recommended as clinical management. The mechanism of interaction was demonstrated in vitro and in vivo. Steric hindrance occurs at the active ASA binding site of COX-1 [20] by metamizole. Nonsteroidal anti-inflammatory agents (nonselective), such as ibuprofen, indomethacin, or naproxen, also inhibit the active ASA binding sites and have the toxic effect on renal function described above. ASA has a half-life of 25 min and is hydrolyzed to salicylic acid. This does not inhibit platelet aggregation. Although the effect of ASA is only short and persists due to the irreversible acetylation of COX-1 of the platelets until it is newly formed, this interaction is present. If the ASA level is very high during the trough of metamizole, there is the possibility of a PAI by ASA. It can be achieved by fast-releasing ASA dosage forms (no enteric-coated tablets) and an intake at the time of the trough level of metamizole (e.g., ASA before breakfast like the thyroid hormones). However, this is controversial in the literature and currently an open question [21]. A recently published study indicates that if treatment with metamizole is unavoidable, the lowest dose effective to relieve pain should be prescribed, and a strict order of intake is crucial [22]. The only case in which the PAI of ASA is not affected is with acetaminophen [23]. 

Another critical combination of angiotensin-converting enzyme inhibitors, such as ramipril with potassium salts, and potassium-sparing diuretics, such as spironolactone and potassium salts, might increase the risk of developing hyperkalemia [24], which is likely to be greater in patients with additional risk factors for the development of hyperkalemia (e.g., renal insufficiency). This interaction is most likely the result of aldosterone suppression in patients receiving angiotensin-converting enzyme inhibitors, which may increase potassium retention.

The combination of NSAIDs and salicylates may enhance the risk of bleeding. Patients at risk for this complication need monitoring. NSAIDs, such as ibuprofen, and possibly other nonselective NSAIDs, may reduce the cardioprotective effects of ASA. It seems prudent to avoid regular and frequent use of ibuprofen in patients receiving ASA for its cardioprotective effects [25,26]. The combination of oral quinolones, such as ciprofloxacin, and oral iron-like iron-II-glycine-sulfate preparations for the substitution of an iron deficit produce a pharmacokinetic interaction, which leads to a reduction in the antibiotic effectiveness of the quinolones in conjunction with polyvalent cations. A chelation between the quinolones and the polyvalent cation is suspected. As a result, the chelate complex formed can no longer be absorbed leading to therapy failure. The effect can be minimized by administering the two medications at different times (2 h before or 6 h after the administration) [27].

The interaction of serotonin/norepinephrine reuptake inhibitors (SNRIs), such as venlafaxine, or selective serotonin reuptake inhibitors (SSRIs), such as citalopram, also has antiplatelet effects [28]. Prior studies found statistically significant associations between concomitant use of SSRIs and ASA and reported an increased risk of gastrointestinal bleeding [29,30,31]. Several possible mechanisms for inhibiting platelet aggregation by SSRIs have been reported, but the primary suspected mechanism is a decrease in platelet serotonin concentrations [32]. Serotonin enhances platelet aggregation. Platelets cannot synthesize serotonin, which is why they take up the serotonin from the blood through a transporter. Inhibition of this uptake by SSRIs is believed to lead to a gradual depletion of platelet serotonin and, consequently, impaired platelet aggregation.

The combination of amiodarone and beta-blockers may result in adverse effects associated with excessive beta-receptor antagonism, such as bradycardia, sinus arrest, AV block, and even cardiac arrest. One randomized controlled trial of 412 patients with implantable cardioverter defibrillators reported a significantly greater rate of bradycardia with amiodarone plus a beta-blocker (metoprolol, carvedilol, or bisoprolol) [33]. Amiodarone is a known inhibitor of human cytochrome P450 activities [34].

Clarithromycin and erythromycin, although not azithromycin, inhibit cytochrome P450 isoenzyme 3A4 (CYP3A4), and inhibition increases the blood concentrations of statins that are metabolized by CYP3A4 [35,36]. Atorvastatin should be limited to a maximum dose of 20 mg/day when used with clarithromycin [37,38]. If this combination is unavoidable, patients must be monitored for atorvastatin toxicity (e.g., muscle aches or pains and renal dysfunction) [39]. Alternatively, rosuvastatin can be used. 

Amiodarone may increase the serum concentration of atorvastatin, and atorvastatin toxicities (e.g., myalgia, liver function test elevations, and rhabdomyolysis) are possible [40,41]. Lower starting and maintenance doses of atorvastatin in patients taking amiodarone must be used. There are currently no studies or case reports documenting an interaction between amiodarone and atorvastatin. However, there are numerous reports describing an interaction between amiodarone and simvastatin, a similar CYP3A4-metabolized statin.

### 3.3. Potential Clinically Relevant Moderate Risk

Corticosteroids (oral), such as prednisolone, may enhance the hypokalemic effect of thiazide diuretics, such as hydrochlorothiazide. Supplementation of and/or a combination with a potassium-sparing diuretic may be necessary with concomitant treatment. It is known that systemic corticosteroids and diuretics (loops and thiazides) cause potassium waste in the kidneys. Concurrent therapy is likely to be additive in this regard. In a large study of drug monitoring in hospitals, the records of 2439 patients on potassium-wasting diuretic therapy were reviewed to identify other factors that affected serum potassium levels. Oral or parenteral prednisone therapy was a significant risk factor for the development of hypokalemia [41].

Simvastatin doses greater than 20 mg should be avoided in patients being treated with amlodipine. The exact mechanism for this potential interaction is uncertain but might involve competition for CYP3A as both simvastatin and amlodipine are substrates of this enzyme. Moreover, lower doses, when indicated, require enhanced monitoring for simvastatin toxicity, that is, signs of rhabdomyolysis [42].

Salicylates double the risk of gastrointestinal bleeding from glucocorticoids, side effects that are very rare when corticoids are given alone. With this combination, it is necessary to treat the patient with a proton pump inhibitor for the prevention of gastrointestinal bleeding [43].

The absorption of levothyroxine can be reduced by the simultaneous oral administration of calcium salts due to a reduced adsorption of levothyroxine to calcium in the gastrointestinal tract. An in vitro study confirmed that in acidic conditions, levothyroxine binds calcium carbonate [44,45]. Therefore, levothyroxine should be taken at least four hours before calcium intake. It is important to monitor for decreased therapeutic effects of thyroid products if an oral calcium supplement is initiated or the dose is increased or for increased effects if an oral calcium supplement is discontinued or the dose is decreased.

Pregabalin may increase the impairment of cognitive and gross motor functions caused by oxycodone. These drugs should only be combined if there are no other treatment options. If the combination is unavoidable, the dose should be kept at a minimum. The initial extended-release dose of oxycodone should be reduced by 50% to 67% in patients already receiving CNS depressants [46].

### 3.4. Notes for Medical Professionals

The use of a benzodiazepine or a Z-substance (zopiclone or zolpidem) in elderly patients should be avoided at all costs because of the risk of falls. In many cases, the drugs melperone and pipamperone are a good treatment alternative.

First-generation statins, such as simvastatin, pravastatin, or fluvastatin, with a short elimination half-life should therefore preferably be taken before bedtime. Other statins, such as rosuvastatin or atorvastatin, can also be taken in the morning.

Drugs that frequently caused problems were the active ingredients of the ATC system: ATC B01 antithrombotic agents; N02 analgesics; A12 minerals; H02 corticosteroids for systemic use; potassium chloride A12BA01; C03D aldosterone antagonists; and other potassium-sparing agents. Ward doctors should watch out for DRPs when patients are prescribed medications from these drug groups.

Our study has limitations. The number of patients included was limited. However, our results show a significant number of drug-related problems. We assumed that these problems would not have been noticed in routine clinical practice and not at any time during the hospital stay. Further investigations should clarify whether patients referred to the hospital by their general practitioners are already taking a problematic drug combination. This study was only carried out in one ward of the vascular surgery department. There is therefore no data from other surgical departments for comparison. Because the study was designed to focus on interactions in vascular surgery patients, the cost savings (e.g., duration of therapy) through the interventions of the clinical pharmacist were not recorded. In addition to this, it is difficult to quantify the savings when an increase in drug safety and an avoidance of DRPs go hand in hand. Through dose-adjustment interventions, evidence of potential interactions and contraindications of drug safety became more obvious. Furthermore, the exchange with the vascular surgeons was difficult at first because the responsibilities were not clearly defined. Medication entry was manual because patient charts were kept in writing. A project is currently underway at Augsburg University Hospital to record medications electronically. The clinical decision support system (CDSS: AiD clinic) is also to be integrated into this system. Therefore, future research could be conducted with larger numbers of patients and in more detail. A pharmacist should also be involved in therapy-relevant decisions in an interdisciplinary team to optimize the medication process and patient treatment. For a targeted use of expertise, pharmacists should focus on special patient groups prone to be affected by medication errors, such as multimorbid, chronically ill, and elderly people [47].

## 4. Conclusions

This study helped to identify problem drug groups and to develop prevention strategies for DRPs in patients with vascular diseases. We showed that special caution is required when prescribing antithrombotic agents, analgesics, minerals, corticosteroids for systemic use, aldosterone antagonists, and other potassium-sparing agents. With attention to the problems of these groups, the vascular surgeon can make drug therapy safer for patients with vascular diseases. Pharmaceutical advice by a clinical pharmacist could also help the vascular surgeon to improve the therapeutic outcome of patients with vascular diseases [48,49].

## Figures and Tables

**Figure 1 medicina-59-00780-f001:**
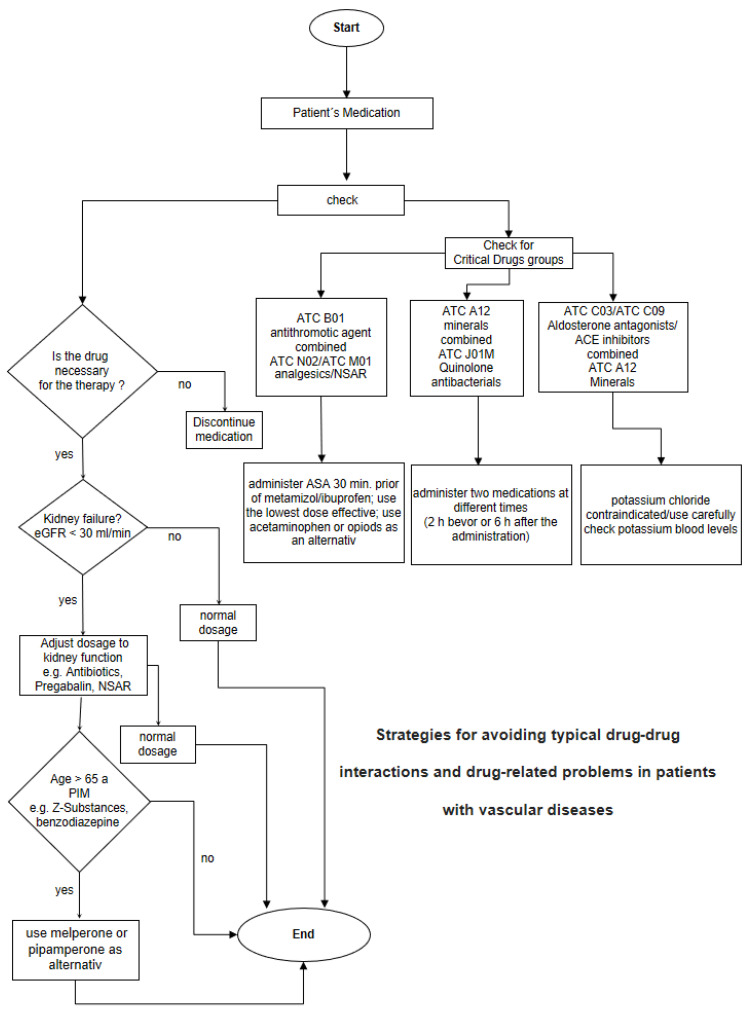
Workflow to systematically avoid the most common DRPs identified in this study.

**Table 1 medicina-59-00780-t001:** Patient characteristics, median (IQR), or *n* (%).

Characteristics	*n* = 96 (IQR)
Gender	
Male	66 (68.8%)
Age (years) mean	73.6 (12.43)
Age range	48–97
Smoker	18 (18.8%)
Former Smoker	13 (13.5%)
Number of DRPs (total)	277
Interactions	111 (2)
Interventions	114 (7)
Notes for ward doctors	96
Kidney failure stage 5: eGFR 5 < 15 mL/min	4 (4.1%)
Kidney failure stage 4: eGFR15–29 mL/min	8 (8.3%)
Kidney failure stage 3: eGFR 30–59 mL/min	30 (31.2%)
Kidney failure stage 2: eGFR 60–89 mL/min	37 (38.5%)
Normal kidney function: eGFR > 90 mL/min	17 (17.7%)
Number of medications	10.5 (4)
Polypharmacy > 5 medications	92 (95.8%)
Main indications of hospitalization (ICD-10):	
Atherosclerosis of the extremity arteries (I70)	49 (51%)
Embolism and thrombosis of the arteries of the lower extremities (I74)	10 (10.4%)
Diabetes mellitus, Type 2 (E11)	10 (9.4%)
Aortic aneurysm and dissection (I71)	7 (7.3%)
Complications due to prostheses, implants, or grafts in the heart and vessels (T82)	5 (5.2%)
Complications indicative of replantation and amputation (T87)	4 (4.2%)
Other aneurysm and other dissection (I72)	3 (3.2%)
Chronic ischemic heart disease (I25)	2 (2.1%)
Skin abscess, boil, and carbuncles (L02)	2 (2.1%)
Occlusion and stenosis of precerebral arteries without resulting cerebral infarction (I65)	1 (1.0%)
Varices of the lower extremities (I83)	1 (1.0%)
Complications of surgery, not elsewhere classified (T81)	1 (1.0%)
Secondary diagnosis:	
Hypertension	66 (68.8%)
Type 1 diabetes	2 (2.1%)
Type 2 diabetes	50 (52.1%)
Peripheral arterial occlusive disease	43 (44.8%)
Cardiac arrhythmia	26 (27.1%)
Renal insufficiency/failure	26 (27.1%)
Hyperlipidemia	23 (24.0%)
Coronary artery disease	22 (22.9%)
Cancer	18 (18.7%)
Pneumonia	12 (12.5%)
COPD	12 (12.5%)
Aneurysm	9 (9.3%)
Heart attack	8 (8.3%)
Heart failure	8 (8.3%)
Anemia	8 (8.3%)
Valvular heart disease	6 (6.3%)
Hypercholesterolemia	6 (6.3%)
Affective disorder	6 (6.3%)
Hypothyroidism	5 (5.2%)
Thrombosis	4 (4.2%)
Gastrointestinal disorders	4 (4.2%)
Embolism	3 (3.1%)
Delirium	3 (3.1%)
Liver disease/insufficiency	3 (3.1%)
TIA	3 (3.1%)
ICD implant	3 (3.1%)
Type 1 diabetes	2 (2.1%)
Coronary artery bypass	2 (2.1%)
Hyperthyroidism	1 (1%)
Blood clotting disorder	1 (1%)
Abdominal surgery	1 (1%)
Aortic valve repair/replacement	1 (1%)

**Table 2 medicina-59-00780-t002:** Interactions based on the Anatomical Therapeutic Chemical Classification System (ATC).

Drug Combinations Involved in Contraindicated/High-Risk Combinations	*n*	Consequences of This Combination
eplerenone C03DA04/potassium chloride A12BA01	4	contraindication ➔ discontinuation of potassium chloride
apixaban B01AF02/enoxaparin B01AB05	1	contraindication ➔ discontinuation of enoxaparin
rivaroxaban B01AX06/enoxaparin B01AB05	1	contraindication ➔ discontinuation of enoxaparin
Total	6	
Drug combinations involved in clinically serious interactions:		
acetylsalicylic acid (ASA) B01AC06/metamizole-natrium N02BB02	24	administer ASA 30 min. prior to metamizole; use the lowest dose effective; use acetaminophen or opioids as an alternative
ramipril C09AA05/potassium chloride A12BA01	4	increased risk of toxic potassium blood levels; use carefully, check potassium blood levels
spironolactone C03DA01/potassium chloride A12BA01	3	increased risk of toxic potassium blood levels;use carefully, check potassium blood levels
ASA B01AC06/ibuprofen M01AE01	3	administer ASA 30 min. prior to ibuprofen
ciprofloxacin J01MA02/iron (II) glycine sulphate complex B03AA01	2	administer ciprofloxacin 2 h before or 6 h after the iron (II) glycine sulphate complex
venlafaxine N06AX16/ASA B01AC06	2	increased risk of gastrointestinal bleeding
citalopram N06AB04/ASA B01AC06	3	increased risk of gastrointestinal bleeding
potassium chloride A12BA01/lisinopril C09AA03	2	increased risk of toxic potassium blood levels; use carefully, check potassium blood levels
acarbose A10BF01/glimepiride A10BB12	1	risk of hypoglycemia
amiodaron C01BD01/metoprolol C07AB02	1	use carefully under ECG controls
atorvastatin C10AA05/valsartan and sacubitril C09DX04	1	increased risk of myopathy and/or rhabdomyolysis use carefully; pay attention to symptoms
atorvastatin C10AA05/amiodaron C01BD01	2	inhibitor of the cyto chrome P450 3A4; increased risk of myopathy and/or rhabdomyolysis; pay attention to symptoms
atorvastatin C10AA05/clarithromycin J01FA09	1	inhibitor of the cyto chrome P450 3A4; increased risk of myopathy and/or rhabdomyolysis; pay attention to symptoms
bisoprolol C07AB07/amiodaron C01BD01	1	use carefully under ECG controls
ciprofloxacin J01MA02/citalopram N06AB04	1	use carefully under ECG controls; check potassium blood levels
clarithromycin J01FA09/atorvastatin C10AA05	1	increased risk of myopathy and/or rhabdomyolysis; pay attention to symptoms
digitoxin C01AA04/torasemide C03CA04	1	increased risk of toxic potassium blood levels; use carefully, check potassium blood levels
duloxetine N06AX21/rivaroxaban B01AX06	1	increased risk of gastrointestinal bleeding; monitor therapy; agents with antiplatelet properties may enhance the anticoagulant effect of rivaroxaban
enoxaparin B01AB05/escitalopram N06AB10	1	increased risk of gastrointestinal bleeding
eplerenone C03DA04/ramipril C09AA05	1	increased risk of toxic potassium blood levels; use carefully, check potassium blood levels
ibuprofen M01AE01/torasemide C03CA04	1	monitoring of diuresis in the first days after starting and stopping NSAIDs
ibuprofen M01AE01/candesartan and hydrochlorothiazide C09DA26	1	monitoring of diuresis in the first days after starting and stopping NSAIDs
ibuprofen M01AE01/hydrochlorothiazide C03AA03	1	monitoring of diuresis in the first days after starting and stopping NSAIDs
ibuprofen M01AE01/valsartan and amlodipine C09DB01	1	monitoring of diuresis in the first days after starting and stopping NSAIDs
lisinopril and diuretics C09BA03/potassium chloride A12BA01	1	increased risk of toxic potassium blood levels; use carefully, check potassium blood levels
melperone N05AD03/citalopram N06AB04	1	use carefully under ECG controls; check potassium blood levels
paroxetine N06AB05/ASA B01AC06	1	increased risk of gastrointestinal bleeding
prednisolone H02AB06/hydrochlorothiazide C03AA03	1	increased risk of toxic potassium blood levels; check potassium blood levels
ramipril C09AA05/eplerenone C03DA04	1	increased risk of toxic potassium blood levels; use carefully, check potassium blood levels
ramipril and diuretics C09BA05/potassium chloride A12BA01	1	increased risk of toxic potassium blood levels; use carefully, check potassium blood levels
ramipril and diuretics C09BA05/ibuprofen M01AE01	1	monitoring of diuresis in the first days after starting and stopping NSAIDs
sertraline N06AB06/ASA B01AC06	2	increased risk of gastrointestinal bleeding; monitor therapy; agents with antiplatelet properties may enhance the anticoagulant effect of ASA
spironolactone C03DA01/ciprofloxacin J01MA02	1	increased risk of toxic potassium blood levels;use carefully, check potassium blood levels
valsartan and diuretics C09DA03/ibuprofen M01AE01	1	monitoring of diuresis in the first days after starting and stopping NSAIDs
azithromycin J01FA10/moxifloxacin J01MA14	1	use carefully under ECG controls; check potassium blood levels
valsartan and sacubitril C09DX04/potassium chloride A12BA01	1	increased risk of toxic potassium blood levels; use carefully, check potassium blood levels
fentanyl N02AB03/clarithromycin J01FA09	1	CYP3A4 inhibitors (strong) may increase the serum concentration of fentanyl, monitoring of opioid side effects (coma, respiratory depression)
levodopa in combination with benserazide N04BA11/melperone N05AD03	1	efficacy of levodopa to be verified in the short term
magnesium aspartate A12CC05/levothyroxine-natrium H03AA01	1	magnesium should be taken no earlier than 2 h after taking levothyroxine
ibuprofen M01AE01/furosemide C03CA01	1	monitoring of diuresis in the first days after starting and stopping NSAIDs
Total:	81	
Drug combinations involved in potential clinically relevant moderate risk combinations:		
prednisolone H02AB06/hydrochlorothiazide C03AA03	3	corticosteroids (systemic) may enhance the hypokalemic effect of thiazide and thiazide-like diuretics; check potassium blood levels
prednisolone H02AB06/torasemide C03CA04	4	corticosteroids (systemic) may enhance the hypokalemic effect of loop diuretics; check potassium blood levels
prednisolone H02AB06/ASA B01AC06	3	ASA may enhance the adverse/toxic effect of corticosteroids; increased risk of gastrointestinal bleeding
amlodipine C08CA01/simvastatin C10AA01	2	inhibitor of the cyto chrome P450 3A4; increased risk of myopathy and/or rhabdomyolysis; pay attention to symptoms
levothyroxine-natrium H03AA01/calcium, combinations with vitamin D and/or other drugs A12AX	2	amlodipine may increase the serum concentration of simvastatin; increased risk for myopathy and/or rhabdomyolysis; pay attention to symptoms
oxycodone N02AA05/pregabalin N03AX16	2	CNS depressants may enhance the CNS depressant effect of oxycodone
ASA B01AC06/ verapamil C08DA01	1	increased risk of gastrointestinal bleeding; calcium channel blockers may enhance the antiplatelet effect of ASA
enoxaparin B01AB05/clopidogrel B01AC04	1	agents with antiplatelet properties may enhance the anticoagulant effect of enoxaparin; monitor closely for symptoms of bleeding
metoprolol C07AB02/amlodipine C08CA01	1	check blood pressure regularly and watch for signs of heart failure
levodopa, decarboxylase inhibitor and COMT inhibitor N04BA03/iron (II) glycine sulfate complex B03AA01	1	iron preparations may decrease the serum concentration of levodopa; separating doses of the agents by 2 or more hours
metamizole-natrium N02BB02/clopidogrel B01AC04	1	may enhance the antiplatelet effect of other agents with antiplatelet properties
metoprolol C07AB02/digitoxin C01AA04	1	bradycardia-causing agents may enhance the bradycardic effect of other bradycardia-causing agents; monitor heart rate and blood pressure more closely
calcium carbonate A12AA04/ciprofloxacin J01MA02	1	iron preparations may decrease the serum concentration of levodopa; separating doses of the agents by 2 or more hours
ramipril and diuretics C09BA05/clopidogrel B01AC04	1	thiazide diuretics may decrease the excretion of calcium; monitor for toxic effects of calcium if a thiazide diuretic is initiated or the dose is increased
Total:	24	

**Table 3 medicina-59-00780-t003:** Interventions of the clinical pharmacist observed by the ward physician in the therapy of patients with vascular diseases.

Intervention	*n*	%
Apply another drug	13	11.4%
Discontinue the drug	41	36.0%
Dose reduction/adjustment	40	35.1%
Apply medication	8	7.0%
Handwritten errors	12	10.5%
Total	114	100%

**Table 4 medicina-59-00780-t004:** Information provided by the clinical pharmacist to the ward physician to optimize the therapy of patients with vascular diseases.

Notes for Medical Professionals	*n*	%
Monitor laboratory values	49	51.0%
Monitor side effects	17	17.7%
Potentially inappropriate medications	18	18.8%
Application time	4	4.2%
Manual transmission errors	8	8.3%
Total	96	100%

## Data Availability

The datasets generated during and/or analyzed during the current study are available from the corresponding author on reasonable request.

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
