# Peer review of "Strategies for Avoiding Typical Drug–Drug Interactions and Drug-Related Problems in Patients with Vascular Diseases"

_medicina, 2023, doi:10.3390/medicina59040780_

Round 1
Reviewer 1 Report
I had the opportunity to review the manuscipt from Schmelzer K et al. The topic is interest, and the results well writen.
However, i have a main concern regarding the methodology of the paper. The authors report that the medications of 1322 patients were reviewed manually in the time period from 11/2017 to 11/2018, but into the clinical decision support system only the medication of 96 patients were entered. Even more, there is not any clear explanation on methods why the whole available population was not used and how the selection of these 96 patients were done. This leads to huge bias and should be changed.
Author Response
"Please see the attachment."

Reviewer 2 Report
In general, I found the topic of the paper interesting. However, the quality of the reporting is not sufficient for publication. The paper should be thoroughly revised and reorganized in order to make it more clear and straightforward to the reader, also providing all the information needed to reproduce the experiment in other settings. English native revision would be also useful.
Please find below some comments and suggestions that I hope will help to improve the paper.
What do you mean with “curve”? The term is used more than once but personally I do not understand the meaning.
Methods
In general, I would suggest to structure the method section with subheadings to make the reporting of the methodology more straightforward for the reader, e.g. Data source, study population, Identification of DRP and DDI, statistical analysis….). The classification used to present results should be also anticipated in the method section. Each category, e.g. “Contraindicated/high-risk combinations”, should be defined. Was the classification based on authors’ judgment, or was taken from other papers/sources?
What are the population selection criteria? i.e. characteristics that patients had to match to be included into the study cohort, and how the characteristics were identified (e.g. hospital diagnoses…) should be clearly defined to make the study reproducible. E.g. “Patients with vascular diseases who need a surgical procedure, that received pharmacological treatment during inpatient care with at least 1 DRP and DDI according to…”
Please, add references for “clinical decision support system (CDSS; AiDKlinik Release 3.5) and an interaction database (Lexicomp® 82 Drug Interactions)”.
Please add a reference for the ATC system. You can also consider citing the website https://www.whocc.no/
Results
For the sake of clarity, I would recommend starting the results section with a statement on the overall number of subjects included in the study cohort, DRP and DDI. (e.g. A total of XX patients with vascular diseases who needed a surgical procedure and received pharmacological treatment during inpatient care were included in the study cohort.)
Table 2 reads “Avoiding Strategie” please correct with a correct english expression. Moreover the first “avoiding strategie” listed in the table is not adequate. Please, correct. I would also suggest adding a column with the possible consequences of DDI before the column with the “Avoiding Strategie”. For instance, this was the case of “duloxetine N06AX21/rivaroxaban B01AX06” for which “agents with antiplatelet properties may enhance the anticoagulant effect of rivaroxaban” is not a strategy for avoiding adverse events rather the adverse event that the DDI can possibly cause.
Figure 1. I find it really difficult to follow the logic behind the flow diagram.
Discussion
The first sentence summarize the result rather that their meaning. I would suggest to avoid reporting results in the discussion.
Please define what TAH is.
The discussion of each interaction found in the study should follow a somewhat similar schema.
I would avoid using ATC codes in the Conclusion section.
Author Response
"Please see the attachment."

Reviewer 3 Report
Dear Author(s),
Thank you for your interesting and valuable study that helped to identify problem drug groups and to develop prevention strategies for drug-related problems in patients with vascular diseases.
Methodology is adequate, results are clearly presented and conclusion are based on the findings obtained. On the other hand, discussion can be shortened in my opinion.
Please avoid redundancy in Table 1. (e.g. type 1 diabetes) and include percentage for T87 primary diagnosis.
In Table 2. please correct mistake - you stated discontinuation of enoxaparin for eplerenone/potassium chloride interaction.
How did you combine CDSS AiDKlinik Release and Lexicomp for interaction check? Which of them did you consider more relevant in case of incongruence between results obtained?
Best regards, Reviewer
Author Response
"Please see the attachment."

Round 2
Reviewer 1 Report
I am satisfied with the provided answer.